# Evidence for spin-to-charge conversion by Rashba coupling in metallic states at the Fe/Ge(111) interface

S. Oyarzún[1,2,3], A.K. Nandy[4], F. Rortais[1,2], J.-C. Rojas-Sánchez[5], M.-T. Dau[1,2], P. Noël[1,2], P. Laczkowski[5], S. Pouget[1,2], H. Okuno[1,2], L. Vila[1,2], C. Vergnaud[1,2], C. Beigné[1,2], A. Marty[1,2], J.-P. Attané[1,2], S. Gambarelli[1,2], J.-M. George[5], H. Jaffrès[5], S. Blügel[4] & M. Jamet[1,2]

The spin–orbit coupling relating the electron spin and momentum allows for spin generation, detection and manipulation. It thus fulfils the three basic functions of the spin field-effect transistor. However, the spin Hall effect in bulk germanium is too weak to produce spin currents, whereas large Rashba effect at Ge(111) surfaces covered with heavy metals could generate spin-polarized currents. The Rashba spin splitting can actually be as large as hundreds of meV. Here we show a giant spin-to-charge conversion in metallic states at the Fe/Ge(111) interface due to the Rashba coupling. We generate very large charge currents by direct spin pumping into the interface states from 20 K to room temperature. The presence of these metallic states at the Fe/Ge(111) interface is demonstrated by first-principles electronic structure calculations. By this, we demonstrate how to take advantage of the spin–orbit coupling for the development of the spin field-effect transistor.

[1] Institut des Nanosciences et l'Energie Atomique et Cryogénie, INAC, Commissariat á aux Energies Alternatives-Univ. Grenoble Alpes, 17 rue des Martyrs, F-38000 Grenoble, France. [2] CEA, INAC, F-38000 Grenoble, France. [3] Departamento de Fisica, CEDENNA, Universidad de Santiago de Chile (USACH), 9170124 Santiago, Chile. [4] Peter Grünberg Institute and Institute for Advanced Simulation, Forschungszentrum Jülich and JARA, 52425 Jülich, Germany. [5] Unité Mixte de Physique, CNRS, Thales, Univ. Paris-Sud, Université Paris-Saclay, 91767, Palaiseau, France. Correspondence and requests for materials should be addressed to S.O. (email: simon.oyarzun@usach.cl) or to M.J. (email: matthieu.jamet@cea.fr).

Nowadays the field of spintronics mainly relies on the exchange coupling in ferromagnetic (FM) materials to generate and detect spin-polarized currents[1]. However, the spin–orbit coupling (SOC) appears now as a very efficient way to complete both operations[2]. Indeed, it has been shown that spin-polarized currents can be generated and detected by the spin Hall effect (SHE) in bulk materials[3] and the Rashba effect at interfaces[4,5]. Although both effects have been first observed in semiconductors[6–9], they are intensively studied into pure heavy metals[10], alloys[11] or at metallic interfaces[12]. Meanwhile the field of semiconductor spintronics is evolving fast to develop the spin field-effect transistor[13] that would represent a real technological breakthrough in microelectronics. Its development in silicon (Si) or germanium (Ge) requires the spin injection, detection and manipulation at room temperature. Although the SHE is very weak in bulk germanium[14], it can be greatly enhanced at the interface between Ge(111) with heavy metals due to the Rashba effect[15]. In ref. 15, a single layer of Pb deposited on Ge(111) leads to a giant spin splitting of 200 meV along the $M\Gamma M$ direction. Moreover, this type of spin-split surface states are believed to be ubiquitous in any kind of interfaces between Ge(111) and other metals[16]. Hence, beyond providing a way to manipulate the electron spin state, these Rashba states could also be used to generate and detect spin currents in germanium at room temperature, which represents a very new paradigm in the field of semiconductor spintronics.

In this study, we demonstrate a giant spin-to-charge conversion at the Fe/Ge(111) interface due to the Rashba effect into spin-split metallic states, despite the relative lightness of Fe and Ge atoms. Spin accumulation is dynamically generated by direct spin pumping into interface states and converted into a two-dimensional (2D) charge current as a result of the Rashba SOC. Ab initio calculations allow us to demonstrate the presence of Rashba states at the Fe/Ge(111) interface and explain, by simple arguments, the large spin-to-charge conversion.

## Results

**Overview**. In open-circuit conditions and for a 200 mW radio frequency (RF) excitation, the generated voltage normalized to the RF power (that is, to $h_{rf}^2$ in $G^2$ where $h_{rf}$ is the radio frequency magnetic field) is 25 μV G$^{-2}$ at 20 K when the Ge(111) substrate is insulating, highlighting the interfacial character of the conversion. It reaches 50 μV G$^{-2}$ at room temperature. This voltage is proportional to the generated charge current and to the resistance of the sample in between the two electrical contacts. For a $2.4 \times 0.4$ mm$^2$ sample, we find a generated charge current of 1 μA G$^{-2}$ constant with temperature. As a comparison, for the same sample dimensions, Rojas-Sánchez et al.[5] found 0.5 μA G$^{-2}$ in NiFe(15 nm)/Ag(10 nm)/Bi(8 nm) by the inverse Edelstein effect (IEE) at the Ag/Bi interface and 0.5 μA G$^{-2}$ in Co(15 nm)/Pt(20 nm) by the inverse SHE in platinum at room temperature[17]. It shows that the spin-to-charge conversion is highly efficient at the Fe/Ge(111) interface. First-principles electronic band structure calculations based on the density functional theory confirm the existence of p–d hybridized metallic states at the Fe/Ge(111) interface. Those states exhibit a Rashba spin splitting up to 50 meV along the $\Gamma K$ direction of the surface Brillouin zone. Moreover, owing to their hybridization with localized Fe d states, spin-up and spin-down subbands are energy split due to an exchange coupling of the order of 180 meV. In this case, the effect of the Rashba SOC together with the exchange coupling leads to an electronic structure with a lack of mirror symmetry with respect to the plane normal to the surface and containing the in-plane Fe magnetization. Considering this particular symmetry breaking of the Fermi surface (FS), we present simple arguments

to explain the large charge current generation by spin pumping into Fe/Ge(111) interface states.

**Sample growth**. For the purpose of this study, we have grown several samples by molecular beam epitaxy (see Methods for details) as follows: Fe/Ge(111), Fe/MgO(5 nm)/Ge(111), Fe/MgO(20 nm)/Ge(111) and Fe/Ge(100), as well as two reference samples Fe/MgO(001) and Fe/SiO$_2$/Si. In Fe/MgO(5 nm)/Ge(111), we reduce the exchange coupling between Ge p states and Fe d states by inserting a 5 nm-thick MgO layer, whereas for a 20 nm-thick MgO spacer, we completely suppress the signal[18]. In Fe/Ge(100), we reduce the spin-to-charge conversion efficiency due to the weak SOC at the Ge(100) surface[19]. The small mismatch ($\approx 1.3\%$) between the lattice constant of Ge and twice the unit cell constant of bcc Fe makes possible the growth of single crystalline Fe films on Ge(100) and Ge(111) surfaces. As shown in Fig. 1a–c, bcc Fe grows epitaxially on Ge(111) with the (111) texture and a 60° lattice rotation with respect to the Ge diamond lattice. The epitaxial relationship is thus Fe(111)[11$\bar{2}$] || Ge(111)[2$\bar{1}\bar{1}$]. The high-resolution scanning transmission electron microscopy image in Fig. 1d confirms the epitaxial growth of Fe on Ge(111) and the very sharp interface. Similarly, bcc Fe grows epitaxially on Ge(100) and X-ray diffraction data give the epitaxial relationship Fe(001)[100] || Ge(001)[100] (see Methods). In both Ge(100) and Ge(111) samples, the Fe lattice parameter deduced from X-ray diffraction data is in agreement with the value for bulk Fe: $a_{Fe} = 2.86(1)$ Å. Based on these results and on ab initio calculations, we propose an ideally sharp Fe/Ge(111) interface in Fig. 1e that minimizes the interface energy. This same atomic configuration is used to perform the first-principles electronic band structure calculations.

**Spin-to-charge conversion**. Spin pumping-voltage measurements are carried out using the experimental configuration shown in Fig. 1f. The results are shown in Fig. 2. At 293 K, we find a large electrical signal of the order of 50 μV G$^{-2}$ for Fe/Ge(111), whereas it drops by a factor 2.5 for Fe/MgO(5 nm)/Ge(111) and almost disappears for Fe/MgO(20 nm)/Ge(111) and Fe/Ge(100). It clearly demonstrates that the Fe/Ge(111) interface plays a crucial role in obtaining huge spin-to-charge conversion: the signal decreases when separating the Fe(111) and Ge(111) surfaces with a thin MgO barrier and even disappears when increasing the MgO thickness or changing the surface symmetry. The MgO layers grown on Ge(111) are polycrystalline as shown by in-situ electron diffraction and contain a high concentration of oxygen vacancies, which favours inelastic tunnelling as emphasized by Fukuma et al.[20]. For transparent tunnel barriers, a non-zero exchange coupling sets in between the ferromagnet (Fe) and the semiconductor interface states (MgO/Ge), which can produce a spin-to-charge conversion. This exchange coupling, even small, may be mediated by direct tunnelling through evanescent waves within the MgO barrier or via localized defects, for example, oxygen vacancies. Such physical process has already been addressed to explain the efficient spin injection in Si or Ge by spin pumping through a tunnel barrier[21–23] and is also at the origin of the voltage signal detected through the 5 nm-thick MgO barrier. On the other hand, for a 20 nm-thick MgO barrier, we do not observe any voltage signal at the FM resonance (FMR) due to the vanishing tunnelling exchange coupling. Moreover, the measurements on the reference samples presented in the Methods allow us to rule out any electrical signal originating from the Fe electrode itself by the inverse SHE[24,25], the IEE at the Fe/MgO interface, anisotropic magnetoresistance (AMR) effects[26] or magnonic charge pumping[27]. Electrical measurements on the

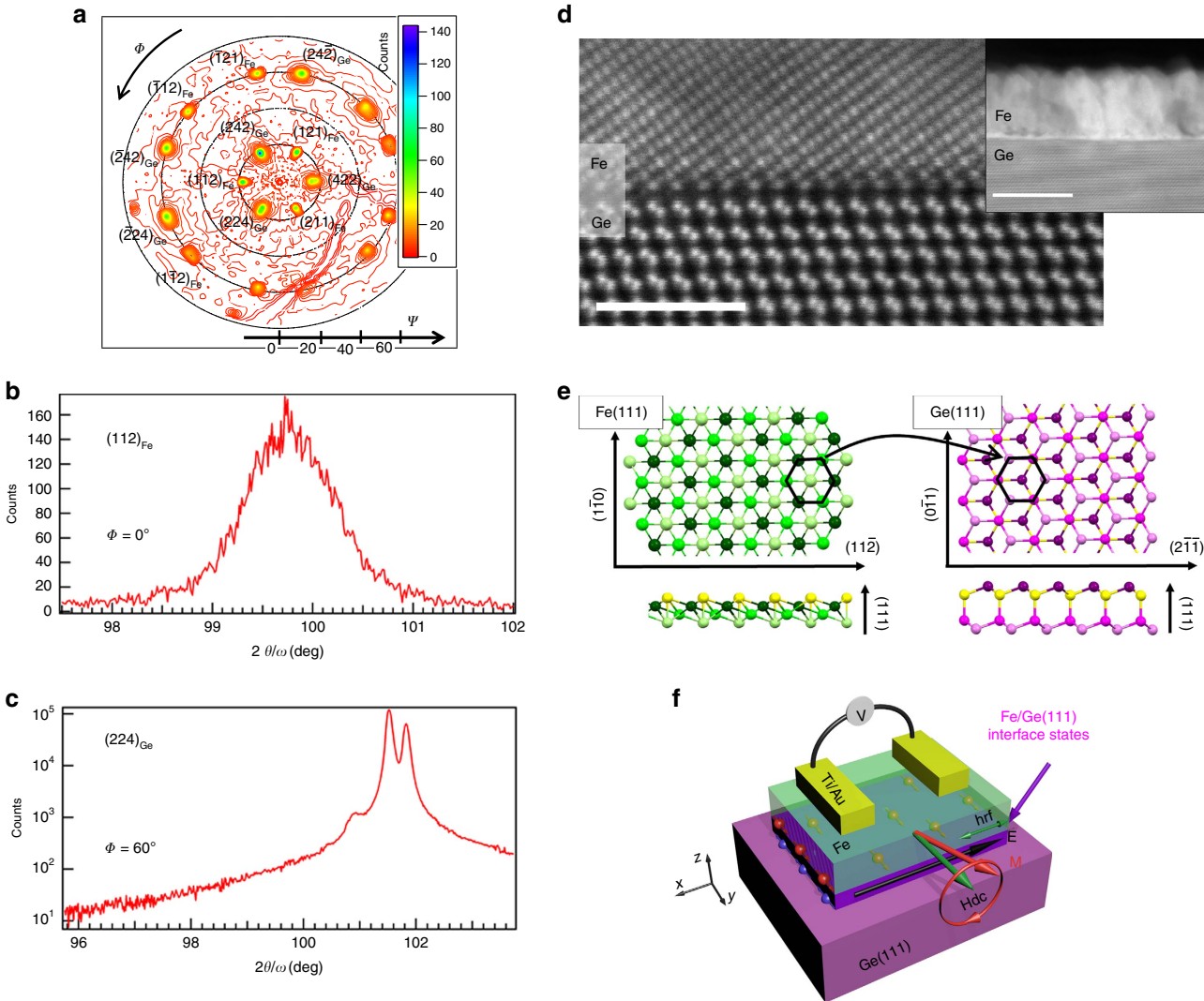

**Figure 1 | Atomic structure of the Fe/Ge(111) interface.** (**a**) X-ray diffraction data for the Fe/Ge(111) sample: (211) pole figure for $2\theta = 99.8°$ with a 3° detector angular acceptance. $\Phi$ and $\Psi$ are the azimuth and declination angles, respectively. (**b**,**c**) $\omega/2\theta$ scan of the Fe (112) and Ge (224) X-ray diffraction lines, respectively. (**d**) High-resolution scanning transmission electron microscopy image of the Fe/Ge(111) interface in cross-section along the $[0\bar{1}1]$ Ge crystallographic axis. Scale bar, 2 nm. Inset: large-scale image of the Fe/Ge(111) interface along the same cristallographic axis. Scale bar, 20 nm. (**e**) Atomic model for the Fe/Ge(111) interface. Light green Fe atoms substitute the dark magenta Ge atoms. (**f**) Sketch of the experimental geometry for spin pumping-transverse voltage measurements. The top Fe layer is contacted using Ti/Au electrodes and the sample is inserted into an X-band cavity working at $f = 9.7$ GHz. The static magnetic field ($H_{DC}$) is applied in the film plane along $y$ and the radiofrequency magnetic field ($h_{rf}$) along $x$.

Fe/Ge(111) sample are shown in Fig. 3a–c. Using the two-probe resistance in Fig. 3a, we can then convert the measured voltage into an electrical current as shown in Fig. 3d. We find a generated current of the order of $1\,\mu A\,G^{-2}$ and constant with temperature. Owing to this large current amplitude, one can merely disregard the inverse SHE in bulk Ge to be the cause of the spin-to-charge conversion because of the relatively small spin Hall angle measured in $n$-type Ge of the order of $\leq 0.1\%$[8,9,14]. Moreover, the electrical signal is constant with temperature, whereas both the Ge(111) substrate and the Fe/Ge(111) Schottky interface become insulating when decreasing the temperature (Fig. 3b,c). This is another strong indication that the conversion mechanism is of pure interface type. The Ge substrate only influences the shape of the electrical signal. As shown in Fig. 3b,c, the Ge substrate becomes conducting and the Fe/Ge(111) Schottky contact more transparent above 200 K. This can be seen in Fig. 3a, where the two-probe resistance starts to drop above 200 K due to the short into the substrate. As a consequence, the

generated electrical current is partly shorted into the substrate above 200 K, which gives an additional asymmetric contribution to the signal as shown in Fig. 3f. At 20 K, the signal is completely symmetric with respect to the resonance magnetic field. Moreover, it changes its sign when the static field is reversed and the signal scales linearly with the RF power (see Fig. 3g).

**Ab initio calculations.** Our first-principles calculations confirm the existence of metallic exchange and spin–orbit split Fe/Ge(111) interface states. Figure 4a,b display the calculated 2D-FS at the Fe/Ge(111) interface within the first Brillouin zone (see Methods for details). The calculations demonstrate the existence of a large density of metallic states in which spin-to-charge conversion might occur. These states exhibit both $p$ and $d$ characters originating from hybridized Ge and Fe states, respectively. In Fig. 4a, without SOC, the FS is sixfold degenerate. The inner blue part shows a snowflake-like structure such as the pristine Ge(111)

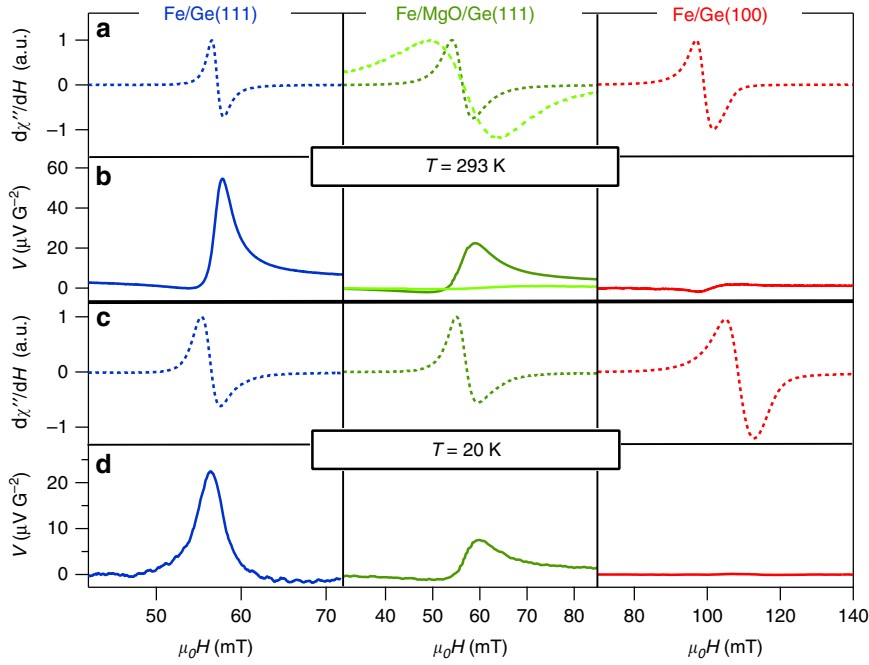

**Figure 2 | FMR and transverse voltage measurements.** The FMR spectrum and transverse voltage are recorded simultaneously for Fe/Ge(111), Fe/MgO/Ge(111) and Fe/Ge(100) at 293 K in **a,b** and 20 K in **c,d**, respectively. Fe/MgO(5 nm)/Ge(111) corresponds to the dark green curve and Fe/MgO(20 nm)/Ge(111) to the light green curve. Fe/MgO(20 nm)/Ge(111) was only measured at room temperature.

surface (not shown here) and thus corresponds mostly to Ge $p$ states with a majority spin character due to the exchange coupling with Fe $d$ states. The loops close to the $K$ points, as the one highlighted in green (for minority spins) and black (for majority spins) along the $\Gamma K_1$ direction, have a dominant $d$ character from Fe states. By including the SOC in Fig. 4b, minority and majority states are no more pure spin states and two main features become visible in the FS. First, the snowflake-like structure at low $k$ values remains but exhibits a strong left–right asymmetry giving rise to a lack of mirror symmetry with respect to the ($yz$) plane containing the Fe magnetization. Then, the SOC modifies the loops close to the $K$ points by introducing a strong spin mixing between majority and minority spin subbands, as well as the same left–right asymmetry. Figure 4c,d display the calculated band structures at the interface along the $\Gamma K_1$ and $\Gamma M$ directions, respectively. Only one-third of the first Brillouin zone corresponding to the snowflake-like structure is represented. At the $\Gamma$ point, the exchange splitting is of the order of 180 meV. The Rashba coupling varies from $\approx 10$ to $\approx 50$ meV, from the $\Gamma$ to the $K_1$ point.

## Discussion

The first mechanism to consider for spin-to-charge conversion at interfaces is the IEE recently demonstrated experimentally in other systems involving Rashba interactions[5,12,28]. The charge current is produced by the spin accumulation generated at the interface between the FM and the non-magnetic 2D material and diffusing out of the FM source. The out-of-equilibrium spin accumulation transforms at the spin–orbit split FS of the non-magnetic 2D material into a charge current, owing to the different electron diffusion coefficients at the majority and minority spin Fermi contours[29]. This can be viewed as an extrinsic contribution. The IEE is well described by a characteristic length, $\lambda_{IEE} = j_C/j_S$, where $j_C$ is the lateral 2D charge current density (in A m$^{-1}$) and $j_S$ is the pumped spin current density (in A m$^{-2}$). This constant

reflects the spin-to-charge conversion efficiency and can be written $\lambda_{IEE} = \frac{\lambda_{SO} \times \tau}{\hbar}$ for a 2D electron gas with Rashba interaction. $\lambda_{SO}$ is the spin–orbit strength (in eV nm) and $\tau$ is the momentum relaxation time also scaling with the spin-flip time $\tau_S$ in the limit of large spin–orbit interaction. As detailed in the Methods, we estimate $\lambda_{IEE} = j_C/j_S \approx 0.13$ nm at room temperature in Fe/Ge(111), which is comparable to the value obtained for Ag/Bi (0.3 nm)[5]. However, *ab initio* calculations show the presence of a substantial exchange interaction within Fe/Ge(111) interface states. It makes possible alternate mechanisms such as direct spin pumping and spin-to-charge conversion into the exchange split interface electron gas. This intrinsic scenario has already been suggested to explain the generation of lateral charge currents by spin pumping at the MgO/Ge interface[21] and at the FM/Topological insulator interface[30,31]. It differs from the IEE by the localization of exchange coupling in the 2D electron gas itself. Moreover, the occurence of a large signal in the Fe/MgO/Ge(111) sample rules out the IEE picture due to the negligible tunnelling transmission through MgO, whereas a small reminiscent exchange coupling is sufficient for the intrinsic scenario. At the FMR of the Fe electrode, the electron spins at the Fe/Ge(111) interface precess following the Fe magnetization under the action of an effective exchange field, which comes from the hybridization of Ge $p$ states with Fe $d$ states. This spin precession creates a steady state out-of-equilibrium spin component along **y**, leading to a chemical potential splitting between majority and minority spins. A net charge current then sets in along the **x** direction due to the lack of mirror symmetry of the electronic density of states with respect to the ($yz$) plane as shown in Fig. 4b. The generated charge current corresponds to the one we detect experimentally. However, an exact estimation of this charge current would require the knowledge of spin accumulation and Rashba coupling in the $k$-space, as well as the electron scattering mechanisms at interfaces, which requires more experimental and theoretical works. It is worth noting that, if we rotate the Fe magnetization

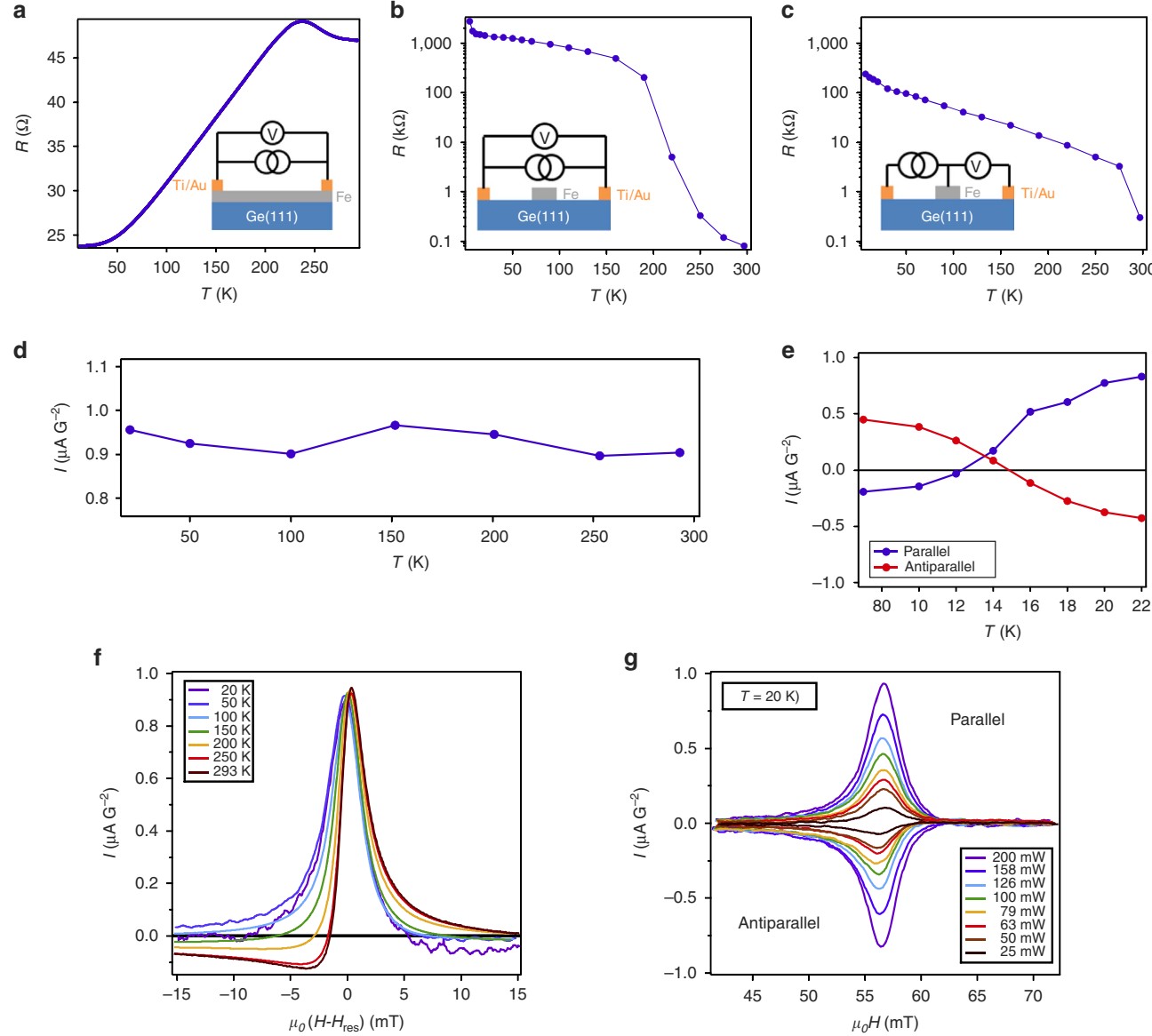

**Figure 3 | Electrical properties of Fe/Ge(111). (a)** Two-probe resistance of the Fe/Ge(111) sample as a function of temperature. **(b,c)** Temperature dependence of the Ge(111) substrate resistance and Fe/Ge(111) Schottky contact resistance, respectively. **(d)** Generated charge current as a function of temperature. The current is defined as: $I = V/(R \times h_{rf}^2)$ where $V$ is the measured transverse voltage at the resonance field, $R$ is the two-probe resistance shown in **a** and $h_{rf} = 0.535$ G is the radiofrequency field corresponding to a 50 mW RF power. The static magnetic field $H_{DC}$ is applied along $y$. **(e)** Low temperature dependence of the generated current for the applied field $H_{DC}$ along $y$ (blue, parallel configuration) and along $-y$ (red, antiparallel configuration). **(f)** Shape evolution of the electrical signal as a function of temperature. A clear asymmetric component appears above 200 K. **(g)** Electrical signal recorded at 20 K as a function of the RF power and the static field direction.

by 90° along **x** (or equivalently, along the $\Gamma M$ direction of the surface Brillouin zone), the generated charge current will be along **y** with a different intensity due to the anisotropy of the 2D electronic band structure of the Fe/Ge(111) interface states. Finally, as shown in Fig. 3e, we observe a sign change of the generated transverse charge current below $T \approx 14$ K. Although the physical origin of this sign change is not yet elucidated, it suggests that this interconversion could be modulated by a gate voltage.

In summary, we have experimentally and theoretically shown giant spin-to-charge conversion at the interface between two light atoms, one of them being germanium, which is compatible with silicon-based technology. This work suggests that spin currents can be generated from charge currents at the single-crystalline Fe/Ge(111) interface. We thus believe that it will motivate future works on spin transport at model interfaces between metal and semiconductor light atoms. Moreover, spin-to-charge conversion at the Fe/Ge(111) interface gives an experimental demonstration in spintronics of spectroscopic predictions. Finally, it shows that SOC in silicon and germanium could be used to generate and detect spin currents, which are the basic operations of the spin field-effect transistor but also to develop interfacial spin-orbit torques for the manipulation of magnetization.

## Methods

**Sample growth and characterization.** Before the sample growth, Ge(111) substrates are degreased 5 min in ultrasonic baths using aceton and isopropyl alcohol, and transferred in deionized water to the growth chamber. Ge surfaces are then annealed up to 850° under ultra-high vacuum conditions, to thermally remove

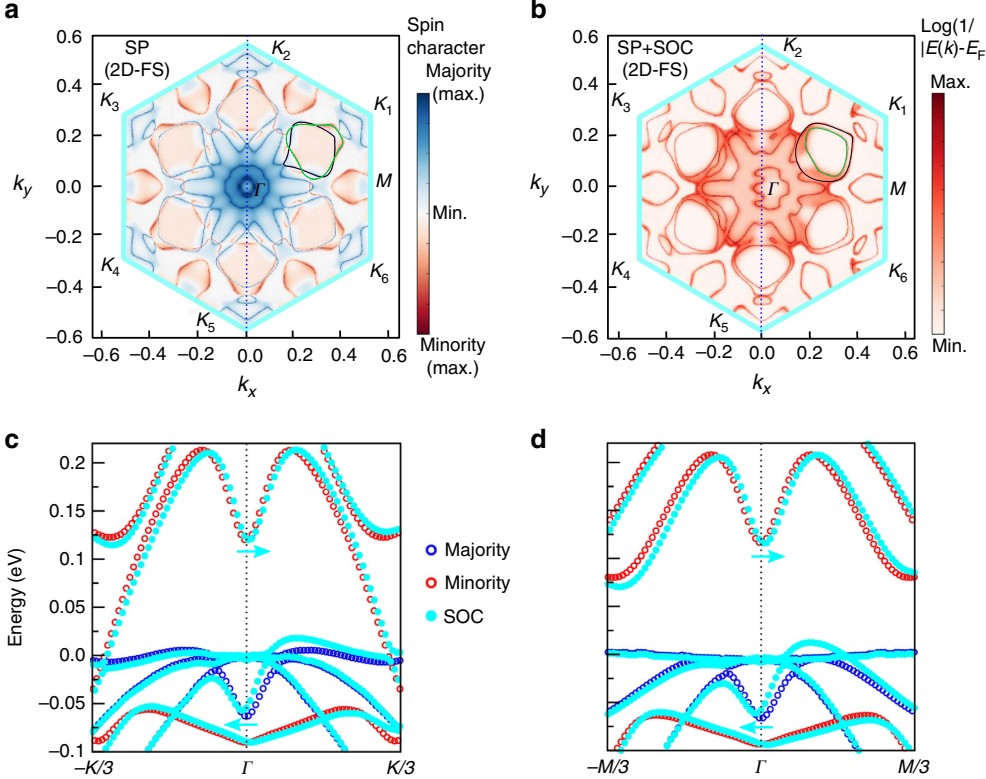

**Figure 4 | Results of *ab initio* calculations.** Two-dimensional Fermi surfaces (2D-FS) in the hexagonal Brillouin zone (**a**) without and (**b**) with SOC for spin-polarized (SP) calculations. The blue dotted line gives the Fe magnetization direction along $y$. In **a**, the blue and red curves correspond to four majority and one minority bands, respectively, crossing the Fermi level. In **b**, the same bands are represented irrespective of the electron spin state. The colour scale relates to the distance of the energy band $E(k)$ from the Fermi level $E_F$ at given $k$ value as $\mathrm{Log}(1/|E(k) - E_F|)$. Hence, max. means a band crossing the Fermi level. Without SOC, the Fermi contours are sixfold degenerate. For convenience, the six equivalent $K$ points are labelled as $K_1$ to $K_6$. Along the $\Gamma K$ direction, majority and minority states are degenerate at the crossing points with zero exchange splitting, which, however, becomes finite due to the SOC and the sixfold degeneracy is broken. This splitting is dominantly contributed by the SOC at the Fe/Ge(111) interface. (**c,d**) Calculated band structures of the Fe/Ge(111) interface states along the $\Gamma K_1$ and $\Gamma M$ directions, respectively. The bands experience a $k$-shift under the action of SOC, in directions indicated by blue arrows, which are opposite for minority and majority bands. Majority and minority states are indicated in black and red dots, respectively. With SOC, majority and minority states lose their meaning and the bands are indicated in light blue.

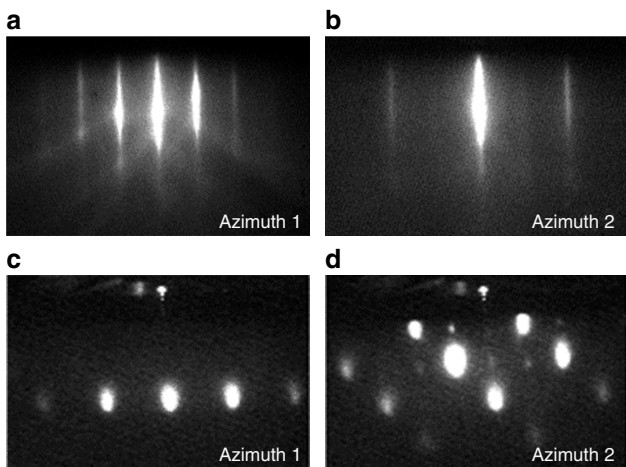

**Figure 5 | RHEED patterns of Ge(111) and Fe(111) surfaces.** The RHEED patterns are shown along two different azimuths of the Ge(111) surface in **a,b** and Fe(111) surface in **c,d**, respectively.

the oxide layer and smooth the surface. The resulting Ge(111) surfaces are atomically flat and not reconstructed as shown by the streaky reflection high-energy electron diffraction (RHEED) patterns in Fig. 5a,b. All the 20 nm-thick Fe layers are grown under ultra-high vacuum conditions with a base pressure

$<5 \times 10^{-10}$ mbar on low-doped $n$-type (111) and (100) Ge substrates with $n \approx 10^{16}$ cm$^{-3}$. The growths are performed at low rates and at room temperature, in order to limit atomic interdiffusion at the Fe/Ge interface[32,33]. As-grown Fe films are single crystalline but their surface is rough as shown by the spotty RHEED patterns in Fig. 5c,d. This surface roughness is a consequence of the low growth temperature. The X-ray diffraction data for the Fe/Ge(100) sample are shown in Fig. 6. They confirm the epitaxial relationship Fe(001)[100] || Ge(001)[100]. Moreover, using the Scherrer equation, the width of the Fe(200) diffraction line yields an iron thickness of 20 nm as expected. The 5 and 20 nm-thick MgO spacer layers are grown on Ge(111) at 300 °C and a low growth rate of 0.025 Å s$^{-1}$. The Ge substrate resistivity is of the order of 1 Ω cm at room temperature. The Fe films are finally protected against oxidation by a 8 nm-thick aluminum capping layer. For the Fe/MgO(001) reference sample, a 20 nm-thick Fe layer is epitaxially grown on MgO(001) at room temperature and a growth rate of 0.5 Å s$^{-1}$. The film is then annealed at 630 °C during 20 min to reach a very high crystalline quality as shown by the streaky RHEED pattern. For the Fe/SiO$_2$/Si reference sample, a 20 nm-thick Fe film is grown at room temperature and a growth rate of 0.5 Å s$^{-1}$. Owing to the amorphous character of the SiO$_2$ substrate, the Fe layer is polycrystalline as shown by the presence of rings in the RHEED pattern.

X-ray measurements are performed with a Panalytical Empyrean diffractometer equipped with a Cobalt anode beam tube ($\lambda_{K\alpha} = 1.789$ Å), a Göbel Mirror and a 2D PIXcel detector used in both zero-dimensional and one-dimensional modes. The cross-sectional electron microscopy specimen was prepared by a focused ion beam. Scanning transmission electron microscopy operation was carried out under a 300 keV acceleration energy using a probe aberration-corrected microscope (Titan Themis FEI).

**Spin pumping and transverse voltage measurements.** After the epitaxial growth, samples are cut into small rectangular pieces of 2.4 mm length and 0.4 mm width. Electrical connections are achieved using aluminum wire bonding. The

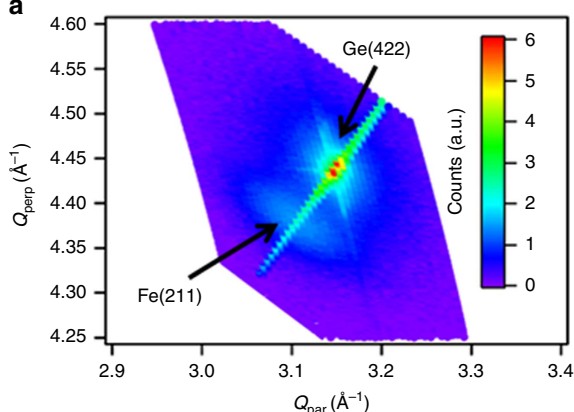

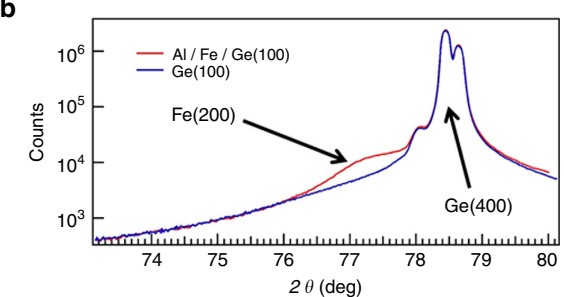

**Figure 6 | X-ray diffraction data for the Fe/Ge(100) sample.**
(**a**) Reciprocal space map of the Fe(211) and Ge(422) reflections. The perpendicular and parallel components of the corresponding momentum transfers $Q_{perp}$ and $Q_{par}$ are along [100] and [011], respectively. (**b**) $\theta/2\theta$ Scans for the Fe/Ge(100) sample and Ge(100) substrate showing the Fe(200) and Ge(400) diffraction lines.

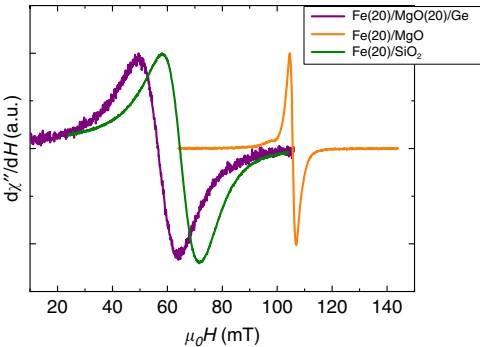

**Figure 7 | FMR spectra of the reference samples.** The spectra are recorded at room temperature under 200 mW for the following: Fe(20 nm)/MgO(20 nm)/Ge(111) substrate; Fe(20 nm)/MgO(001) substrate; and Fe(20 nm)/SiO₂/Si. Owing to the single crystalline character of Fe(20 nm)/MgO(001), the magneto-crystalline anisotropy shifts the resonance field to higher values.

samples are finally glued on a sample holder made of printed circuits and placed in the centre of a cylindrical X-band resonator cavity ($f \approx 9.7$ GHz, $TE_{011}$ mode). The spin pumping and transverse voltage measurements are obtained on two different samples with similar results. In some other samples, we observe a strong symmetric voltage at the resonance field with the same sign for $H_{DC}$ applied along $y$ and $-y$. We attribute this signal to the Seebeck effect in germanium. When the radio-frequency power is not equally distributed over the length of the sample, a temperature gradient appears between both ends of the sample, which gives rise to a large Seebeck voltage. It can be almost removed by precisely positioning the

sample inside the cavity to put the maximum radiofrequency power in the middle of the sample.

The FMR spectra and voltage measurements on the reference samples Fe/MgO(001) and Fe/SiO₂/Si are shown in Fig. 7 and in Fig. 8, respectively. In both samples, we could not detect any voltage signal at room temperature.

To estimate the characteristic IEE length $\lambda_{IEE}$, we consider the Fe/Ge(111) system as a simple bilayer system where the spin accumulation level is maintained constant in Fe by the spin pumping mechanism and spin flips occur into the Fe/Ge(111) interface states. The Fe/Ge(111) metallic interface states are not an ideal spin sink but their spin absorption efficiency nonetheless scales with the change of the FMR spectra linewidth, whereby the effect of the spin backflow process[34] is involved in the change of the Fe damping parameter $\alpha$. To take this spin backflow process into account, we define an effective spin mixing conductance[35]: $g_{eff}^{\uparrow\downarrow} = \frac{4\pi M_{eff} t_F}{g \mu_B}(\alpha - \alpha_{Fe\ bulk})$. $M_{eff}$ is the effective saturation magnetization defined as: $4\pi M_{eff} = 4\pi M_S + H_{u\perp}$. $M_S$ is the saturation magnetization of the Fe electrode and $H_{u\perp}$ corresponds to the perpendicular uniaxial anisotropy field. $t_F = 20$ nm is the thickness of the Fe electrode, $g_{eff}$ is the Landé factor and $\mu_B$ the Bohr magneton. $\alpha$ (resp. $\alpha_{Fe\ bulk}$) is the damping factor of Fe/Ge(111) (resp. bulk Fe). The Landé factor $g_{eff}$, the effective saturation magnetization of Fe $M_{eff}$ and the Gilbert damping factor $\alpha$ are deduced from broadband FMR measurements at room temperature. The measurements are summarized in Fig. 9. The frequency dependence of the resonance field is fitted using the following equation:

$$\left(\frac{\omega}{\gamma}\right)^2 = H_{res}(4\pi M_{eff} + H_{res}) \tag{1}$$

where $\omega = 2\pi f$ and $\gamma = g_{eff}\mu_B/\hbar$ is the electron gyromagnetic ratio. In the same way, the frequency dependence of the peak-to-peak linewidth is given by:

$$\Delta H_{pp} = \Delta H_0 + \frac{2}{\sqrt{3}}\frac{\omega}{\gamma}\alpha \tag{2}$$

where the $\Delta H_0$ term accounts for the frequency-independent contributions due to inhomogeneities in the FM layer. From the fits, we find the following: $\mu_0 M_{eff} \approx 1.83$ T, $g_{eff} = 2.1$ and $\alpha = 0.0032$. Using $\alpha_{Fe\ bulk} = 0.0018$ (ref. 36), we obtain $g_{eff}^{\uparrow\downarrow} \approx 2.6 \times 10^{19}$ m⁻². The vertical spin current density transferred from the Fe layer to the Fe/Ge(111) interface states is then given by[37]:

$$j_S = \frac{g_{eff}^{\uparrow\downarrow}\gamma^2\hbar h_{rf}^2}{8\pi\alpha^2}\left[\frac{4\pi M_S\gamma + \sqrt{(4\pi M_S\gamma)^2 + 4\omega^2}}{(4\pi M_S\gamma)^2 + 4\omega^2}\right]\frac{2e}{\hbar} \tag{3}$$

Using $f = 9.7$ GHz, $\omega = 2\pi f$ and $h_{rf} = 0.535$ G, we find $j_S \approx 1.73 \times 10^7$ A m⁻². The spin-to-charge conversion only occurs into the Fe/Ge(111) interface states, as no signal could be detected in the reference samples Fe/MgO(001) and Fe/SiO₂/Si. The generated charge current density $j_C$ writes: $j_C = V/(Rw)$ where $V \approx 46\ \mu$V is the measured voltage at room temperature, $R \approx 51\ \Omega$ is the sample resistance between the two voltage probes and $w = 0.4$ mm is the sample width. We find the generated charge current: $j_C = 2.257 \times 10^{-3}$ A m⁻¹ and derive $\lambda_{IEE} = j_C/j_S \approx 0.13$ nm.

**Ab initio calculations.** The electronic structure calculations were performed employing the full-potential linearized augmented plane wave method[38,39] as implemented in the FLEUR code (http://www.flapw.de). A mixed functional was used for the exchange-correlation potential based on the approximation by Perdew[40] where all the gradient terms were set to zero within the Ge muffin-tin spheres. In this way, both Fe and Ge were treated with optimal functionals: the Fe atoms were better described within the generalized gradient approximation while the lattice constant of Ge was well reproduced in the local-density approximation (LDA). Generalized gradient approximation overestimates the lattice constant of Ge by 2.5%, whereas in LDA it is only 0.6% smaller compared with the experimental lattice constant 5.657 Å. However, it is well known that LDA produces a metallic ground state for Ge. To model a semiconducting ground state, we follow the suggestion in ref. 41 using an LDA + U scheme that optimizes the band-gap energy at experimental lattice constant. The on-site Coulomb parameters $U_{dd} = 6$ eV applied on the $3d$ electrons and $U_{pp} = -3$ eV on the $4p$ electrons are used to optimize the position of $d$-like and $p$-like states, respectively. Now, instead of a metallic ground state, this approach opens up indirect and direct band gap energies of $\sim 0.4$ and $0.5$ eV at the $\Gamma$ and $L$ points in the Brillouin zone (BZ), respectively, in close vicinity to the experimental values, although the energies are slightly underestimated in comparison with the experimental values, for example, $0.66$ eV at the $L$ point. Our asymmetric film consist of 10 layers of Fe in fcc stacking on 23 layers of the Ge(111) substrate terminated with hydrogen atoms, to simulate the semi-infinite substrate. The total energy of different Fe-Ge stackings were calculated. The stacking of Fe-Ge with lowest energy is consistent with the experimental results and the surface unit cell comprises one atom per layer. The distances between the different layers were relaxed until the forces on the atoms comprising the top three Fe layers, as well as the Fe/Ge interface and sub-interface layers were smaller than 1 mRyd per a.u. For the structurally optimized system we have calculated the electronic structure and the FS properties with and without SOC. For the convergence of self-consistently determined quantities, the plane wave cutoff energy and the **k**-points in the 2D BZ were checked carefully. The calculations were totally converged with and without SOC using $\mathbf{k}_{||} = 256$ in the full

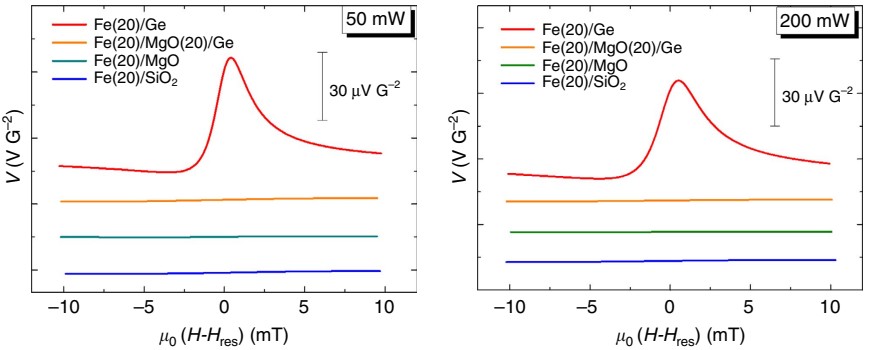

**Figure 8 | Transverse voltage measurements on reference samples.** Room-temperature transverse voltage (normalized to the RF power) generated by spin-to-charge conversion at the FMR of the Fe electrode at two different RF powers 50 and 200 mW in the following: Fe(20 nm)/Ge(111); Fe(20 nm)/MgO(20 nm)/Ge(111) substrate; Fe(20 nm)/MgO(001) substrate; and Fe(20 nm)/SiO₂/Si.

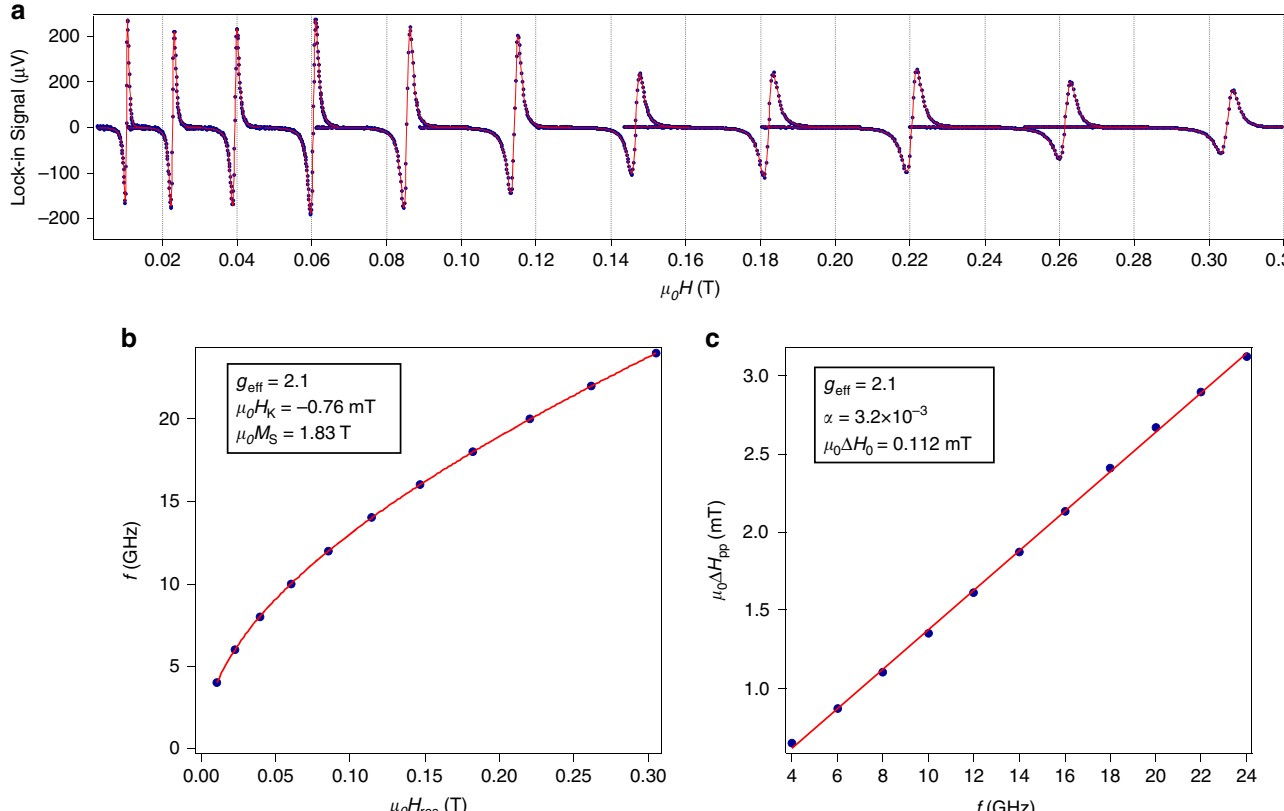

**Figure 9 | Broadband FMR measurements at room temperature.** The static magnetic field is applied in the film plane. (**a**) FMR lines recorded at different excitation frequencies from 4 GHz (left) to 24 GHz (right). Black dots are experimental data and solid lines are fits using a Lorentzian shape for the absorption line. The resonance fields and peak-to-peak linewidths are extracted from the fits. (**b**) Resonance field $\mu_0 H_{res}$ as a function of the excitation frequency. Black dots are experimental data and solid lines are fits using equation (1). (**c**) Peak-to-peak linewidth $\Delta H_{pp}$ as a function of the excitation frequency. Black dots are experimental data and the solid line is a fit using equation (2).

2D BZ with a charge-density cutoff parameter of 15.0 per a.u. and a plane-wave cutoff parameter of 3.8 per a.u. to expand the linearized augmented plane wave basis functions. The 2D-FS was calculated for $\mathbf{k}_{\parallel} = 2,304$ in the full BZ using converged charge density and a linear interpolation scheme was used to calculate the $\mathbf{k}_{\parallel}$ points close to the Fermi level by a threshold value of ∼1 meV.

**Data availability.** The experimental data that support the findings of this study are available from the corresponding authors upon reasonable request. The results of *ab initio* calculations are also available from AKN (email: Ashis.Nandy@physics.uu.se) upon reasonable request.

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

## Acknowledgements

We acknowledge the financial support from the French National Research Agency through the ANR projects SiGeSPIN ANR-13-BS10-0002 and SOspin ANR-13-BS10-0005. They also acknowledge Nicolas Mollard for the preparation of electron microscopy specimens. Professor Aurélien Manchon and Dr Nicolas Reyren are also acknowledged for fruitful scientific discussions.

## Author contributions

S.O. and M.J. supervised the project. M.J. proposed the study. S.O., J.-C.R.-S., M.-T.D. and P.N. performed the spin pumping measurements with the help of S.G., A.K.N. and S.B. performed and interpreted the *ab initio* calculations. S.O., F.R., C.V. and C.B. performed the electrical measurements on the Fe/Ge(111) sample. M.J. and A.M. performed the epitaxial growth of Fe/Ge samples. S.P. performed and analysed the X-ray diffraction data. H.O. made the transmission electron microscopy observations. H.J., S.B. and M.J. proposed the physical model for the spin-to-charge conversion. S.O., A.K.N. and M.J. wrote the paper with the comments of P.L., J.-M.G., J.-P.A. and L.V.
