## [Peer Review File · Nature Communications]

REVIEWERS' COMMENTS:

Reviewer #1 (Remarks to the Author):

The authors did an excellent job in addressing all issues raised. While I still think that the explanation is rather speculative, it cannot be ruled out. In any case, the results will attract the interest of the community and I think that the manuscript can be published in the present form.

Reviewer #2 (Remarks to the Author):

I recommend publication of this paper in Nature Communications. I believe the additional experiments the authors have done has crossed the appropriate threshold. It is unfortunate that it is much easier to conceptually change the experimental geometry than it is to do in practice. I still think it would be valuable to see measurements for additional geometries, but I accept the authors' pleadings that they would be an inappropriate difficulty. The additional work the authors have done convinces me that it is not mere sloth that has kept them from the measurements I would like.

I think the authors' arguments have become strong enough that their results will be of sufficient interest to the community to justify their publication.

Reviewer #3 (Remarks to the Author):

After carefully reading the response of the authors to the previous reviewer criticism, I think that most points raised by myself and the other reviewers have been satisfactorily addressed. Thus my conclusion remains pretty much unchanged from my first review and I support publication of this article.

I just have two remaining minor issues, which I leave at the discretion of the authors.

I think it is good that authors nor reference the early work on spin Hall effects in Ge on page 4, but I think they should so already in the opening paragraph of the introduction, i.e., when they discuss that first experimental observations were observed in semiconductors (together with Refs. 5 and 6), and when they mention that spin Hall effects are very weak in Ge (together with Ref. 11).

When the authors use MgO to suppress the spintransport across the interface from Fe to Ge, they may want to cite Appl. Phys. Lett. vol. 96, 022502 (2010), which as far as I know is the first publication that a thin MgO layer can effectively suppress the interfacial spin current from spin pumping.

point-by-point response to the issues raised by the referees

Referee #3

“I think it is good that authors nor reference the early work on spin Hall effects in Ge on page 4, but I think they should so already in the opening paragraph of the introduction, i.e., when they discuss that first experimental observations were observed in semiconductors (together with Refs. 5 and 6), and when they mention that spin Hall effects are very weak in Ge (together with Ref. 11).”

Answer

As requested by Referee #3, we have moved ref. “Chazalviel, J.-N. & Solomon, I., Experimental Evidence of the Anomalous Hall Effect in a Nonmagnetic Semiconductor, Phys. Rev. Lett. **29**, 1676 (1972).” and ref. “Chazalviel, J.-N., Spin-dependent Hall effect in semiconductors, Phys. Rev. B **11**, 3918 (1975).” in the opening paragraph of the introduction. They are now ref. 8 and 9 in the revised manuscript. The same references are cited when we mention that the spin Hall effect is weak in Ge together with ref. 14 as also requested by the Referee #3.

“When the authors use MgO to suppress the spintransport across the interface from Fe to Ge, they may want to cite Appl. Phys. Lett. vol. 96, 022502 (2010), which as far as I know is the first publication that a thin MgO layer can effectively suppress the interfacial spin current from spin pumping.”

Answer

As suggested by Referee #3, we also added the following reference:

Mosendz, O., Pearson, J. E., Fradin, F. Y., Bader, S. D., Ho_mann, A., Suppression of spin-pumping by a MgO tunnel-barrier. Appl. Phys. Lett. 96, 022502 (2010).

dealing with the suppression of the interfacial spin current from spin pumping. It is now Ref. 18 cited in page 3 of the revised manuscript.